# Basic emergency obstetric and newborn care service availability and readiness in Nepal: Analysis of the 2015 Nepal Health Facility Survey

Kiran Acharya[1], Raj Kumar Subedi[2], Sushma Dahal[3], Rajendra Karkee[4] *

1 New ERA, RudramatiMarga, Kalopul, Kathmandu, Nepal, 2 Bhaskar Tejshree Memorial Foundation, Kathmandu, Nepal, 3 School of Public Health, Georgia State University, Atlanta, GA, United States of America, 4 School of Public Health and Community Medicine, BP Koirala Institute of Health Sciences, Dharan, Nepal

* rkarkee@gmail.com

**Data Availability Statement:** Data is available through the DHS program (https://www.dhsprogram.com/data/dataset_admin/login_main.cfm).

## Abstract

### Background

Achieving maternal and newborn related Sustainable Development Goals targets is challenging for Nepal, mainly due to poor quality of maternity services. In this context, we aim to assess the Basic Emergency Obstetric and Newborn Care (BEmONC) service availability and readiness in health facilities in Nepal by analyzing data from Nepal Health Facility Survey (NHFS), 2015.

### Methods

We utilized cross-sectional data from the nationally representative NHFS, 2015. Service availability was measured by seven signal functions of BEmONC, and service readiness by the availability and functioning of supportive items categorized into three domains: staff and guidelines, diagnostic equipment, and basic medicine and commodities. We used the World Health Organization's service availability and readiness indicators to estimate the readiness scores. We performed a multiple linear regression to identify important factors in the readiness of the health facilities to provide BEmONC services.

### Results

The BEmONC service readiness score was significantly higher in public hospitals compared with private hospitals and peripheral public health facilities. Significant factors associated with service readiness score were the facility type (14.69 points higher in public hospitals, P<0.001), number of service delivery staff (2.49 points increase per each additional delivery staff, P<0.001), the service hours (4.89 points higher in facilities offering 24-hour services, P = 0.01) and status of periodic review of maternal and newborn deaths (4.88 points higher in facilities that conducted periodic review, P = 0.043).

**Funding:** The authors received no specific grant from any funding agency in public, commercial, or not-for-profit sectors.

**Competing interests:** The authors have declared that no competing interests exist.

## Conclusions

These findings suggest that BEmONC services in Nepal could be improved by increasing the number of service delivery staff, expanding service hours to 24-hours a day, and conducting periodic review of maternal and newborn deaths at health facilities, mainly in the peripheral public health facilities. The private hospitals need to be encouraged for BEmONC service readiness.

## Introduction

Globally between 2000 to 2017, maternal mortality ratio (MMR) and neonatal mortality rate (NMR) have dropped substantially by about 38% and 50%, respectively [1, 2]. Despite this reduction, the number of women and newborns dying each day due to preventable causes related to pregnancy, childbirth, and postpartum is huge, reaching as high as 810 for women [3] and 7000 for neonates in 2018 [4]. Studies have also revealed that with a quality emergency obstetric and newborn care in place, we could prevent as many as 60% of maternal deaths and 85% of intrapartum related deaths per year [5].

Nepal has made a remarkable progress in increasing the utilization of maternal health services. According to the Nepal Demographic and Health Survey (NDHS) 2016 [6], seven out of ten women made at least four antenatal care visits, and six of ten women delivered their babies in a health facility. Similarly, Nepal made substantial progress in reducing its pregnancy related mortality ratios from 543 per 100000 live births in 1996 to 259 per 100000 live births in 2016 [6]. The neonatal mortality rate was stagnant, with a rate of 33 per 1000 live birth both in 2006 and 2011. However, in 2016 the neonatal mortality rate declined rapidly to 21 per 1000 live births [6]. Owing to these achievements, Nepal was one of the exemplary countries to meet Millennium Development Goals (MDGs) related to child survival (MDG-4), and maternal health (MDG-5), and its success stories have been globally commended [7].

Nevertheless, the neonatal mortality rate (NMR) accounts for more than half (54%) of the deaths in children younger than five years in Nepal [6]. In addition, the progress in maternal newborn and child health indicators is unequal across different socio-economic levels in the country and has remained stagnant for the last decade [6]. As a signatory nation to United Nations Sustainable Development Goals (UN-SDGs), the Government of Nepal is devoted to achieving the SDGs targets through the provision of equitable, respectful and quality maternity services [8]. Equitable and high-quality health system can prevent half of all maternal deaths and is key factor that will contribute to progress toward the SDG targets in maternal health in low resource settings [9].

Timely access to emergency obstetric and newborn care services are needed for the delivery of life saving interventions that treat major causes of maternal and newborn mortality and morbidity [10]. These services have been differentiated based on the level of care into basic and comprehensive emergency obstetric and newborn care services. The BEmONC services include seven signal functions [11]. It has been reported that maternity services and health workers' skills do not meet best practice standard. Recent data from the Maternal and Perinatal Death Surveillance in Nepal indicates that most maternal deaths occur in the postpartum period and of those deaths, 48% occurred within 48 hours. Further, accessing quality maternity services remains challenging in Nepal, especially for women living in the rural mountainous and hill regions [12].

Pregnancy complications are unpredictable and can occur in 15% of pregnancies in general [13]. Thus, health facilities offering normal vaginal delivery services should be prepared to provide seven basic emergency and obstetric newborn care (BEmONC) signal functions—to manage health complications when they occur [14]. These signal functions can be used to assess availability of safe delivery and newborn care services [15]. In this context, it is imperative to track the status of health facility readiness to meet the maternal and newborn related targets of SDGs. Thus, we aim to assess the basic emergency obstetric and newborn care service availability and readiness among health facilities prior to the COVID-19 pandemic in Nepal using nationally representative data from NHFS, 2015.

## Materials and methods

### Study design

This study analyzed the data from the NHFS conducted in 2015. The 2015 NHFS is the first nationally representative cross-sectional health facility survey in Nepal to assess the availability and readiness of providing basic and essential health services in the health facilities including components of BEmONC. This survey was conducted by New ERA, a local research firm, in partnership with the Nepal Ministry of Health and Population. Technical assistance for survey implementation was provided by ICF International under the Demographic and Health Survey program with financial support from United States Agency for International Development and UK Department for International Development [16]. We used freely available public domain survey data provided after detaching all the identifier information. The survey was approved by the Ethics Committee of the ICF International in the USA and by the Nepal Health Research Council in Nepal. During the survey, informed consent was requested and obtained from the participants before the interview.

### Sample and sampling procedure

NHFS, 2015 sampled a total of 1000 health facilities using a random stratified sampling technique. After removing 8 observations due to duplicate of health facilities and 29 observations of health facilities that either refused to participate, were closed on the interview days or could not be reached because of poor infrastructure, a total of 963 health facilities participated in the survey. The 2015 NHFS was designed to provide national-level representative results by facility type and by management authority (ownership of facility), the facility are also representative at the national and provincial level [17]. For the purpose of this study, we included only those health facilities (n = 457) that provide delivery and newborn care services. The details of survey methodology including sampling procedure were reported elsewhere [16].

### Data collection

The 2015 NHFS data collection took place from April 20 to November 5, 2015, using five types of survey instruments: Facility Inventory Questionnaire, Health Provider Questionnaire, Exit Interview Questionnaires, Questionnaire for Health Facility Operations and Management Committees, and Observation Protocols for antenatal care, family planning, and service for sick children. For this study, we analyzed variables from the Facility Inventory Questionnaire and one variable on staff training from the Health Provider Questionnaire.

Data related to service availability and readiness for BEmONC were provided by the health facility in-charge, the most senior health worker, or most knowledgeable health worker available at the time of data collection. The Facility Inventory Questionnaire and Provider Questionnaire were administered using tablet computers [18]. Eighty-nine trained interviewers

who were either nursing graduates or public health graduates were employed as field data collectors and were supervised throughout the survey by eight quality assurance officers. The monitoring of data quality was done simultaneously as data was collected. Survey teams were also in close contact with NHFS central office at New ERA through field visits by the key survey team.

## Measurement of variables

The outcome variables are 'BEmONC services availability' and 'BEmONC services readiness.' These variables were chosen based on the WHO service availability and readiness (SARA) manual by the three domains of tracer indicators, and each domain consists of a set of tracer items [15]. 'BEmONC services availability' was defined as the physical presence of the services related to the provision of BEmONC services. This was measured based on the seven signal functions of delivery and newborn care services carried out by health facility within past 3 months preceding the survey: parenteral administration of antibiotic, parenteral administration of oxytocin, assisted vaginal delivery, parenteral administration of anticonvulsants, manual removal of retained products of conception, manual removal of placenta, and neonatal resuscitation. The survey items related to these signal functions whether or not the facility offered these services. If the facility did offer the service, the provider responding on behalf of the health facility was then asked if the service or services have been provided at least once during the prior 3 months. These seven signal functions are measures of the responsiveness of health services to the key obstetric complications at the basic levels, which resemble roughly to the health center level and the level of the first-referral hospital, respectively [16].

Likewise, we defined 'BEmONC services readiness' as the preparedness of the facility to provide BEmONC services. This was assessed based on the availability and functioning of supportive items categorized into three domains: staff and guidelines (2 indicators), essential equipment and supplies (14 indicators), and basic medicine and commodities (11 indicators). The list of tracer items of each domain is provided in S1 Table.

The independent variables included in the regression analysis were: facility type (hospitals or peripheral), ownership of the facility (private or public), location of the facility (rural or urban, and by province), ecological region (Mountain, Hill and Terai), external supervision in the facility in the past 4 months (occurred and not occurred), reviews of maternal or newborn deaths (reviewed and not reviewed), system of determining and reviewing clients' opinion (reviewed and not reviewed), quality assurance performed at least once a year (performed and not performed), duty schedule or call list for 24-hour staff assignment (Yes and No),the number of delivery service staff and the number of beds per facility. The peripheral health facilities include primary health care centers, health posts, and urban health centers. The location of facilities (rural/urban) was not available in the datasets. We have classified them using the global positioning system [19]. These key variables were taken from the available literature [20, 21] and were used to demonstrate the availability or readiness of the health facility to provide BEmONC services.

## Statistical analysis

We used a weighted additive procedure to measure the BEmONC service readiness from the three domains. This procedure involves assigning equal weights to each domain, and adjusting for the "variation in the number of indicators within each domain so that the weight of the indicator is inversely proportional to the number of indicators in the domain" where a facility obtains a total score—that is, the sum of all indicators were standardized to have a maximum score of 100. Equal weighting is the most spontaneous approach to generate a composite

measurement compared with other frequently used weighting patterns [19–22]. The measurement procedure of the readiness score is displayed in the S1 Table.

The numbers of facilities of each type are weighted or adjusted so that each type's contribution to the total is proportionate to the actual distribution of health facilities in the country. So, we used facility weights prior to analysis, to restore representativeness of the health facilities sampled. We also accounted for the NHFS complex sampling design when estimating Standard Errors (SEs) through the use of Stata's 'svy' commands. All analyses were done using Stata version 15.0 (Stata Corp, College Texas).

We summarized our descriptive analysis for continuous variables using mean (SD) for normally distributed variables and median (IQR) for the variables that showed skewed distribution. Likewise, for the categorical variables, we calculated proportions. We fitted simple and multiple linear regression models to assess the association between BEmONC services readiness score and independent variables. As the aim was to fit a final model that predicts the association, the objective criterion-based method was used to decide on the variables to be included in the multiple regression models. We used variance inflation factor to check for multicollinearity among the independent variables. A p-value of <0.05 was considered indicative of statistically significant association.

## Results

### General characteristics of surveyed facilities

Table 1 shows a summary of the general characteristics of the health facilities that provide BEmONC services. More than 80% of the health facilities were peripheral facilities and about 90% of the health facilities were public. Approximately 53.1% of the facilities were located in rural areas, while 46.9% were in urban areas. Three-fifths (60.4%) of the facilities were located in the hilly region, less than 10% of the health facilities were located in province 2, with the highest proportion of facilities located in Bagmati province (18.0%). About one in eight health facilities had the system of determining and reviewing clients' opinions (12%) and regularly reviewed maternal and newborn deaths that occurred within the health facility (13.0%). More than 70% of the health facilities had external supervision that occurred within the last four months. Overall, the number of delivery service staff per facility was low, with a median (IQR) of 2 (1–3).

### BEmONC services availability by types of facility

Table 2 shows the distribution of the availability of the seven signal functions for BEmONC by type of health facility and the ownership of health facility. The overall availability of these seven signal functions was higher among hospitals than peripheral health facilities. Regardless of facility type, the majority of the facilities reported high availability of parenteral administration of oxytocin (85.8%) followed by manual removal of placenta (42.8%), parenteral administration of antibiotic (40.7%), neonatal resuscitation (36.8%), and the manual removal of retained products of conception (33%). Less than 20% of the facilities reported to have assisted vaginal delivery (16.1%), and parenteral administration of anticonvulsants services (10.0%). With the exception of parenteral administration of oxytocin, there is significant association between signal functions and type and ownership of facility.

### BEmONC services readiness

Table 3 provides information on BEmONC service readiness in three domains: staff and guidelines, essential equipment and supplies, and medicines and commodities. Approximately 35% of health facilities reported having at least one staff who had received refresher training

**Table 1. General characteristics of health facilities (n = 457).**

| Characteristics | n = 457 (%) |
|---|---|
| **Facility type** | |
| Hospitals | 65 (14.2) |
| Peripheral facilities | 392 (85.8) |
| **Ownership of facility** | |
| Private | 45 (9.8) |
| Public | 413 (90.2) |
| **Location of facility** | |
| Rural | 243 (53.1) |
| Urban | 215 (46.9) |
| **Ecological region** | |
| Mountain | 67 (14.8) |
| Hill | 276 (60.4) |
| Terai | 114 (24.8) |
| **Province** | |
| Province 1 | 78 (16.9) |
| Province 2 | 39 (8.6) |
| Bagmati province | 82 (18.0) |
| Gandaki province | 66 (14.4) |
| Lumbini province | 63 (13.8) |
| Karnali province | 62 (13.5) |
| Sudhurpaschim province | 67 (14.7) |
| **Duty schedule for 24 hours** | |
| Yes | 107 (23.4) |
| No | 350 (76.6) |
| **Quality assurance** | |
| Performed | 93 (20.3) |
| Not Performed | 365 (79.7) |
| **Maternal/newborn deaths** | |
| Reviewed | 59 (13.0) |
| Not reviewed | 398 (87.0) |
| **Clients' opinions** | |
| Reviewed | 55 (12.0) |
| Not reviewed | 402 (88.0) |
| **External supervision in the last 4 months** | |
| Occurred | 328 (71.8) |
| Not Occurred | 129 (28.2) |
| **Number of delivery service staffs per facility** | |
| Median (IQR) | 2 (1,3) |
| **Number of delivery beds per facility** | |
| Mean (SD) | 1.26 (0.74) |

on delivery care, such as training in skilled birth attendant services or active management of the third stage of labor in the past year. Similarly, about one-fourth (21.8%) of health facilities reported having recommended guidelines related to delivery and newborn care (e.g., Nepal Medical Standard Volume III or Reproductive Health Clinical Protocol).

The majority (96.3%) of facilities reported to have at least one dedicated bed for safe delivery followed by the availability of sterilization equipment (92.9%), sterile gloves (92.5%),

**Table 2. BEmONC services availability by type of health facility.**

| BEmONC services | Total n (%) | Facility type | | p value* | Ownership of facility | | p value* |
|---|---|---|---|---|---|---|---|
| | | Hospitals n (%) | Peripheral facilities n (%) | | Public n (%) | Private n (%) | |
| Total | 457 | 65 | 392 | | 413 | 45 | |
| Parenteral administration of antibiotic | 186 (40.7) | 50 (77.4) | 136 (34.7) | <0.001 | 154 (37.3) | 32 (72.3) | <0.001 |
| Parenteral administration of oxytocin | 392 (85.8) | 56 (85.6) | 337 (85.8) | >0.05 | 357(86.6) | 35 (79.0) | 0.33 |
| Parenteral administration of anticonvulsants | 46 (10.0) | 24 (36.7) | 22 (5.5) | <0.001 | 32 (7.7) | 14 (31.2) | <0.001 |
| Assisted vaginal delivery | 75 (16.1) | 29 (44.8) | 44 (11.3) | <0.001 | 58 (14.1) | 15 (34.2) | <0.001 |
| Manual removal of placenta | 196 (42.8) | 41 (62.3) | 155 (39.6) | <0.001 | 172 (41.7) | 24 (53.6) | 0.008 |
| Manual removal of retained products of conception | 151 (33.0) | 37 (56.3) | 114 (29.1) | <0.001 | 131 (31.8) | 20 (43.9) | 0.004 |
| Neonatal resuscitation | 168 (36.8) | 39 (59.4) | 127 (33.0) | <0.001 | 147 (35.6) | 22 (48.0) | 0.009 |

*p-value from chi-square tests of association.

**Table 3. BEmONC services readiness among surveyed health facilities (n = 457).**

| Indicators | n (%, 95% CI) |
|---|---|
| **Staff and guidelines** | 100 (21.8, 17.3–27.1) |
| Presence of guidelines | 100 (21.8, 17.3–27.1) |
| Availability of trained staff | 161 (35.1, 30.1–40.5) |
| **Equipment and supplies** | |
| Emergency transport | 285 (62.3, 56.6–67.6) |
| Sterilization equipment | 425 (92.9, 89.1–95.4) |
| Examination light | 278 (60.7, 55.0–66.2) |
| Delivery pack | 423 (92.4, 88.3–95.2) |
| Suction apparatus | 283 (62.0, 56.6–67.1) |
| Manual vacuum extractor | 95 (20.7, 17.6–24.3) |
| Vacuum aspirator or D&C kit | 88 (19.2, 16.1–22.6) |
| Neonatal bag and mask | 379 (82.8, 77.5–87.1) |
| Delivery bed | 441 (96.3, 93.5–98.0) |
| Partograph | 366 (80.0, 75.0–84.2) |
| Gloves | 423 (92.5, 89.0–95.0) |
| Infant weighing scale | 411 (89.9, 85.6–93.0) |
| Blood pressure apparatus | 385 (84.1, 79.3–88.0) |
| Soap and running water or alcohol-based hand disinfectant | 340 (74.3, 68.9–79.1) |
| **Medicines and commodities** | |
| Essential medicines for delivery | |
| Injectable antibiotic | 187 (40.9, 35.5–46.6) |
| Injectable uterotonic | 403 (88.2, 83.9–91.4) |
| Injectable magnesium sulfate | 330 (72.2, 66.6–77.1) |
| Injectable diazepam | 78 (17.0, 14.0–20.4) |
| Intravenous fluids | 413 (90.3, 86.4–93.2) |
| Skin disinfectant | 418 (91.4, 87.4–94.2) |
| Essential medicines for new-born | |
| Antibiotic eye ointment | 180 (39.5, 33.9–45.3) |
| Chlorhexidine gel | 265 (58.0, 52.1–63.6) |
| Injectable gentamicin | 342 (74.8, 69.2–79.6) |
| Injectable ceftriaxone | 55 (12.0, 10.0–14.4) |
| Amoxicillin suspension | 117 (25.7, 21.2–30.7) |

delivery packs (92.4%), neonatal bag and masks for resuscitation (82.8), and a blank parto-graph (80.0%). About two-thirds of the facilities had emergency transport (62.3%), suction apparatuses (62.0%), and examination lights (60.7%). However, less than a quarter of health facilities reported having a manual vacuum extractor (20.7%) and vacuum aspirator or dilata-tion and curettage (D&C) kit (19.2%).

Intravenous fluids with infusion set and skin antiseptic were the most extensively available of the medicines considered essential for delivery care (90.3% and 91.4%, respectively). Simi-larly, the majority of health facilities had injectable uterotonic (88.2%) followed by injectable magnesium sulfate (72.2%). Conversely, only around two-fifths of health facilities had an injectable antibiotic (40.9%). Less than one-fifth of the health facilities had injectable diazepam (17.0%). Regarding essential medicines for newborn care, three-fourths of health facilities had injectable gentamicin (74.8%), and nearly six in ten (58.0%) had chlorhexidine gel. Two-fifths of the health facilities had tetracycline eye ointment (40.0%). However, a minority of health facilities had amoxicillin suspension (25.7%), or ceftriaxone powder (12.0%) for injection. More than 80% facilities had infant weighing scale and blood pressure apparatuses. About one fourth (25.7%) of the facilities had soap and running water or alcohol-based hand disinfectant in the service site.

Fig 1 shows the distribution of health facilities by BEmONC readiness score. Fig 2 shows the overall readiness score to provide BEmONC services as well as the readiness level of each of the three domains of readiness. The overall mean readiness score of the health facility to

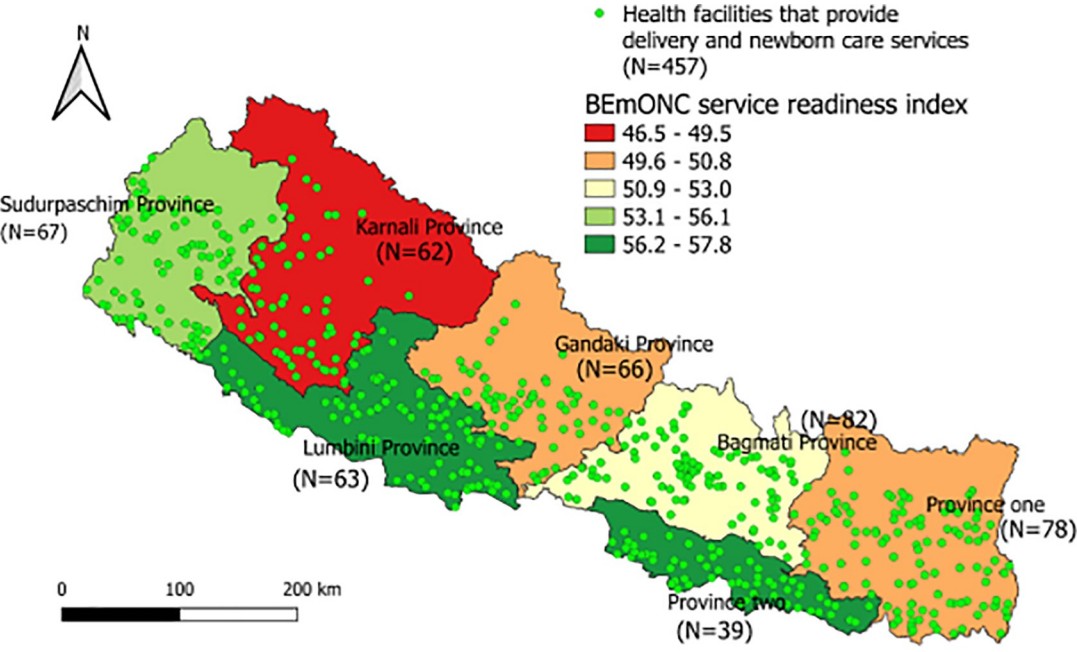

Map was plotted using QGIS from the freely available GPS datasets of the health facilities from www.dhsprogram.com. Shapefile was used from the open access. Dot indicates the number of facilities providing delivery and newborn care service (N=457) and the legend color indicates the BEmONC service readiness index.

**Fig 1. BEmONC service readiness index in health facilities by province-wise.**

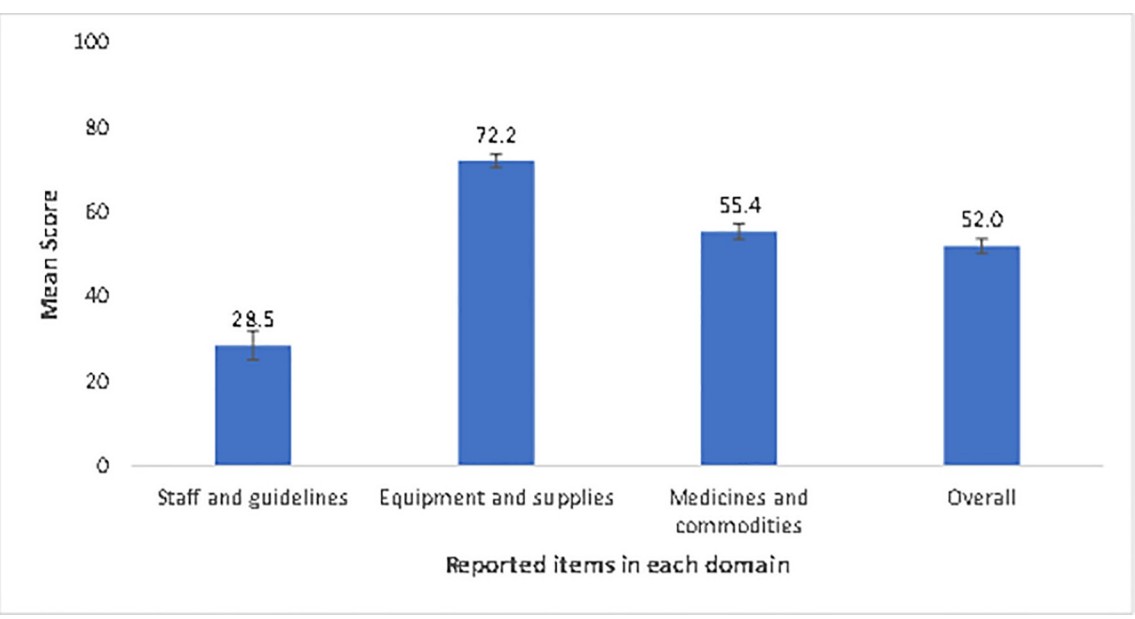

**Fig 2. Mean score of the three domains of readiness to provide BEmONCservices.**

provide BEmONC services was 52.02 (SD = 15.65). The overall readiness score differed significantly according to the type of facility, with a higher score among public hospitals and compared with private hospitals and peripheral public health facilities (p<0.001) (Fig 3).

Table 4 shows the results of the simple and multiplelinearregression models. Collectively, the variables included in the model explained 22.1% of the variation in BEmONC service readiness (R squared = 0.221). The results of multiple linear regression analysis showed that the public facilities have higher readiness score (14.69 points, p<0.001) compared to the private facilities. Similarly, health facilities from Lumbini province had higher readiness scores (7.55 points, P = 0.014) compared to those from Province 1. The readiness of the health facilities to provide BEmONC services was 4.89 points higher (P = 0.01) at facilities having 24 hour staffing than those that do not offer 24-hour services. The facilities that reported regular reviewing of maternal and newborn deaths had 4.88 points higher readiness (P = 0.043) compared to facilities that did not report reviewing maternal and newborn deaths. The service readiness in facilities increased 2.49 points (P<0.001) for each additional delivery staff person working at the facility.

## Discussion

In this study, we assessed the availability and readiness of BEmONC services using data from NHFS, 2015. The overall availability of the seven signal functions and BEmONC service readiness score was significantly higher in public hospitals compared with private hospitals and peripheral public health facilities. Factors associated with service readiness were the ownership of the facility, the number of service delivery staff, the service hours and status of periodic review of maternal and neonatal deaths.

The mean readiness score in this study is higher compared with estimates from some other low-income countries. For instance, in Tanzania, only 29.5% of the health facilities were ready to provide BEmONC services [20]. Another study from Kenya showed that only about 16% of health facilities in rural areas were ready to provide BEmONC services [23]. Our mean readiness score for BEmONC services is higher than the score of overall health service readiness for

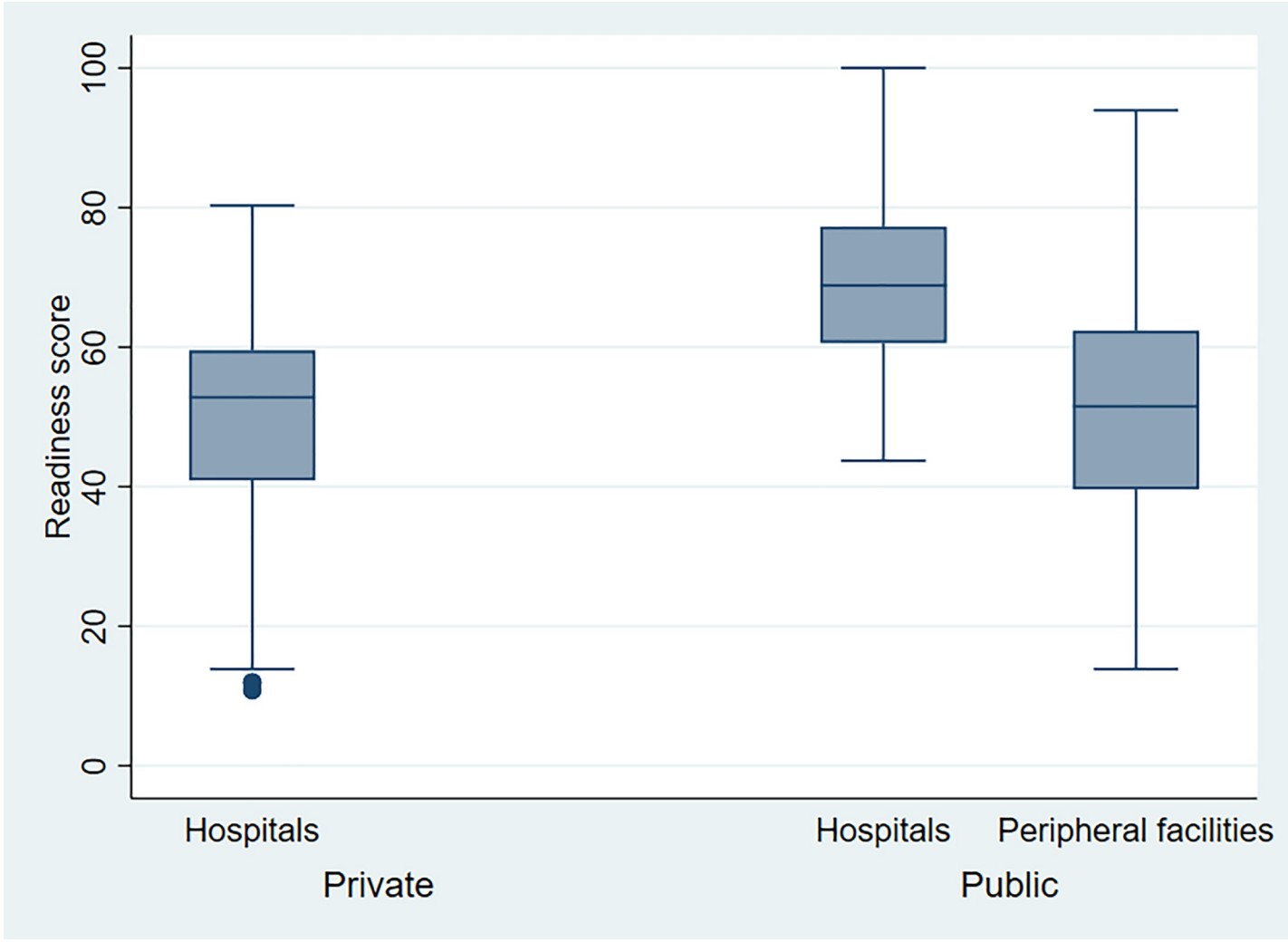

**Fig 3. Overall readiness score of BEmONC by the type of facility.**

Nepal. According to a study that reviewed the service readiness of health facilities in ten countries, including Nepal, the mean health service readiness score was less than 70% for all ten countries, with 40% of them, including Nepal having a readiness score of less than 50% (41% in Bangladesh, 41% in Uganda, 44% in Nepal and 48% in Tanzania) [24].

Evidence suggests that the availability of the seven signal functions for BEmONC is vital to decrease MMR and NMR [2, 25]. In this study, the private facilities had significantly higher availability of all the signal functions compared with the public facilities, with the exception of parental administration of the oxytocin, which was not statistically significant. This is consistent with one study conducted in Pakistan [26]. In this study, both public and private hospitals reported significantly greater availability of the signal functions than peripheral health facilities. This finding is consistent with studies conducted in Tanzania [20] and Haiti [27]. Similarly, the availability of parental administration of anticonvulsants is inadequate in peripheral health facilities of Nepal as has been reported from other similar low resource countries [20, 28]. One of the reasons for the low availability of the seven signal functions in peripheral health facilities in Nepal is the long procurement process of medicines and equipment through a system called the integrated logistics management information system. Further, findings from the

**Table 4. Factors associated with BEmONC services readiness (n = 447).**

| Variable | Unadjusted | | | Adjusted | | |
|---|---|---|---|---|---|---|
| | Coefficient[a] | 95% CI | P-value | Coefficient[a] | 95% CI | P-value |
| **Facility type** | | | | | | |
| Hospitals | ref. | | | ref. | | |
| Peripheral facilities | -4.44 | (-7.91, -0.97) | 0.012 | -4.62 | (-9.56,0.32) | 0.067 |
| **Ownership of facility** | | | | | | |
| Private | ref. | | | ref. | | |
| Public | 2.94 | (-1.30,7.18) | 0.174 | 14.69 | (9.25,20.14) | <0.001 |
| **Location of facility** | | | | | | |
| Rural | ref. | | | ref. | | |
| Urban | 5.17 | (1.75,8.59) | 0.003 | 3.12 | (-0.82,7.07) | 0.121 |
| **Ecological region** | | | | | | |
| Mountain | ref. | | | ref. | | |
| Hill | -2.44 | (-7.33, 2.45) | 0.328 | -4.18 | (-8.76,0.40) | 0.074 |
| Terai | 4.56 | (-0.46, 9.59) | 0.075 | -1.80 | (-7.20,3.61) | 0.514 |
| **Province** | | | | | | |
| Province 1 | ref. | | | ref. | | |
| Province 2 | 6.85 | (0.33,13.37) | 0.039 | 4.95 | (2.03,11.93) | 0.165 |
| Bagmati province | 2.29 | (-3.84,8.41) | 0.464 | 2.97 | (-2.49,8.42) | 0.286 |
| Gandaki province | -0.24 | (-6.56,6.07) | 0.940 | 0.74 | (-5.48,6.96) | 0.815 |
| Lumbini province | 7.96** | (2.14,13.78) | 0.007 | 7.55 | (1.54,13.57) | 0.014 |
| Karnali province | -3.41 | (-10.24,3.41) | 0.326 | -1.92 | (-8.84, 4.98) | 0.584 |
| Sudhurpaschim province | 3.67 | (-2.15,9.50) | 0.216 | 2.05 | (-3.48,7.58) | 0.467 |
| **Duty schedule for 24 hours** | | | | | | |
| No | ref. | | | ref. | | |
| Yes | 8.65 | (5.44,11.86) | <0.001 | 4.89 | (1.16,8.62) | 0.010 |
| **Quality assurance** | | | | | | |
| Not Performed | ref. | | | ref. | | |
| Performed | 3.94 | (-0.12,8.01) | 0.057 | 1.36 | (-2.34,5.06) | 0.470 |
| **Maternal/newborn deaths** | | | | | | |
| Not reviewed | ref. | | | ref. | | |
| Reviewed | 8.29 | (3.68,12.90) | <0.001 | 4.88 | (0.16,6.93) | 0.043 |
| **Clients' opinions** | | | | | | |
| Not reviewed | ref. | | | ref. | | |
| Reviewed | 4.63 | (0.20,9.07) | 0.040 | 2.26 | (-2.42,6.93) | 0.344 |
| **External supervision in the last 4 months** | | | | | | |
| Not Occurred | ref. | | | ref. | | |
| Occurred | 2.52 | (-1.53,6.56) | 0.222 | -0.39 | (-4.25,3.46) | 0.842 |
| **Number of delivery service staffs per facility (as continuous variable)** | 3.58 | (2.46,4.71) | 0.000 | 2.49 | (1.31,3.68) | <0.001 |
| **Number of delivery beds per facility (as continuous variable)** | 3.11 | (1.08,5.14) | 0.003 | 1.35 | (-0.60–3.31) | 0.173 |

[a]Unstandardized Coefficient that represents a change in the outcome on average for a unit change in the independent variable.

Comprehensive Health facility Survey, 2015 also indicated that less than one percent of health facilities had reported tracer medicines available in the facility on the day of the survey [16]. It is also important to note that we found only 16% of health facilities have the availability of assisted vaginal delivery which could be due to the rising trend of the Cesarean Section in Nepal [29].

We found significant differences in the BEmONC service readiness scores across ownership of health facilities, with a higher readiness score in public hospitals compared to private hospitals. This might be due to the significant investment in public hospitals to enable them to provide basic and comprehensive reproductive health services [30, 31]. Similarly, the readiness was lower in peripheral facilities compared to hospitals and facilities in the urban areas of Nepal had higher readiness score compared to rural areas. This is consistent with findings from other countries, such as Kenya [23] and Madagascar [32] where studies have shown a disparity in BEmONC readiness levels between urban and rural health facilities, with rural health facilities having low levels of readiness of BEmONC services. Peripheral facilities are the first level healthcare sites, particularly in the rural areas in Nepal. Moreover, Nepal has a higher number of peripheral facilities, mainly located in rural areas compared to other types of health facilities [33]. Therefore, a lower level of readiness of these front-line health facilities reflects a major challenge in the delivery of adequate BEmONC services in rural areas.

We found that essential equipment and supplies such as sterilization equipment, delivery pack, delivery bed, and gloves were available in the majority of health facilities. A higher proportion of health facilities had essential medicines needed for safe delivery than the essential medicines for the newborns. For instance, the majority of the hospitals had injectable uterotonic, injectable magnesium sulfate, intravenous fluids, and skin disinfectants, while only two-fifth of the health facilities had antibiotic gel ointment available. The overall mean readiness score for medicines and commodities was only 55.4%. This is consistent with a study that found the low availability of generic medicines across the WHO regions. The study analyzed data from 45 surveys in 36 countries and found a low availability of 15 generic medicines across WHO regions that ranged from 29.4% in Africa to 38.3% in South East Asia and 54.4% in the Americas. Findings from this study suggest that the availability of medicines in private health facilities could be improved, with the availability of medicines available in these facilities ranging from 50.1% in the Western Pacific Region to 54.6% in Africa and 75.1% in South-East Asia [34]. Although Nepal aims to provide these essential safe delivery and newborn medicines free of cost in public health facilities under the national free health care program, the unavailability of these medicines in these public facilities is a significant barrier in having access to essential health care services.

The overall availability of the seven signal functions was higher in hospitals compared to peripheral facilities. Although classified as the BEmONC facilities, most of the health facilities at peripheral facilities in Nepal lack a number of services related to emergency obstetric and newborn care. Because of this, many women bypass the community-level health facilities to deliver their babies in large hospitals [35–37]. This practice of bypassing rural health facilities in expectation of better services in urban hospitals is seen in other countries as well, such as India [38], Bhutan [39], Uganda [40]. Improving the availability of equipment and drugs in the primary health care facilities in Nepal, especially in rural areas, is necessary for improving the overall quality of care of maternal newborn health care services [41].

In this study, the readiness score was significantly associated with the number of service delivery staff, type of health facilities, provision of 24 hour BEmONC services, and provision of periodic review of maternal and newborn deaths. This indicates the need for improvement in the infrastructure, review processes, as well as human resource aspects of the health institutions to improve the overall BEmONC service readiness. A study in the Democratic Republic of Congo highlighted the importance of periodic audits of maternal and neonatal deaths along with improving the availability of drugs, equipment, and human resources [42]. It was only after 2015 that Nepal adopted the policy of Maternal Perinatal Death Surveillance and Response (MPDSR) and the implementation of periodic death audits is yet to be measured [43]. Similarly, as a large chunk of health expenditure in Nepal is out of pocket spending [44],

a lower level of readiness in private sectors compared to the public is a significant concern. Thus, it signifies the huge responsibility of the government to improve the quality of health posts, which are located in almost all remote villages of the country as well as to regulate the private health facilities to improve their quality.

Quality assurance is an essential component of any health care, including obstetric and newborn care [45]. Our study shows that only about one in four health facilities had recommended guidelines for delivery and newborn care. We also found that only 12% of the health facilities had the provision of determining and reviewing the clients' opinions. Similarly, only 13% of the health facilities reviewed the maternal and newborn deaths that occurred in the health facility. This indicates a low involvement of health facilities in having adequate review systems in place in Nepal. Less than two-fifths (35.1%) of health facilities reported having at least one staff who had received refresher training on safe delivery, such as training on SBA services or active management of third stage of labor (AMTSL) within 24 months prior to the survey. Similarly, about one-fourth (21.8%) of health facilities reported having recommended guidelines related to delivery and newborn care (e.g., Nepal Medical Standard Volume III or Reproductive Health Clinical Protocol).

Quality of care is fundamental to universal health coverage, and a poor quality of service can undermine the consumer's trust [46]. Evidence suggests that low and middle-income countries are lagging behind in terms of quality of care in areas such as diagnostic accuracy and adherence to clinical practice guidelines [47]. Still many women and children die or develop lifelong disabilities due to poor quality of care in the health facility. Especially in developing countries, where there is high maternal and neonatal mortality, quality assurance is of urgent importance to achieve the mortality reduction targets of mothers and newborns [48]. Our study revealed that the median number of safe delivery service staff in Nepal was 2.0 which is a small number to provide quality emergency obstetric care to the clients. Only 35.1% of the health facilities had adequately trained staff available. A study conducted in Tanzania revealed that the median number of delivery staff was 3 (ranging from 2 to 6) [20]. In addition to these low numbers of safe delivery staff in the health facilities, some studies have even revealed a skill gap among those staff, because of inadequate clinical exposure [49]. Having an adequate number of skilled staff is essential to providing quality health care services, and the shortage of health workers could derail the achievement of health-related goals [50]. A number of studies have identified the reasons behind the low quality of and access to basic emergency obstetric and neonatal services. Mahato and colleagues [51] suggested that we could improve the quality of care of BEmONC facilities by addressing the determinants such as improving the availability of drugs and equipment, provision of trained health staff, improving the health worker-patient relationship, as well as improving physical access to health facilities.

The findings from this study will be helpful for identifying the potential areas of improvement in maternal and neonatal health services to save the lives of many mothers and newborns. Health service planning can be done to address the gaps identified by this study so that the country can accelerate towards the goal of maternal and neonatal related SDGs. However, it should be noted that improving the readiness and quality of health services is not a monolithic formula to reduce maternal and newborn deaths. Efforts should also be directed towards improving the socio-economic status of the people. A study in Haiti, a country with one of the world's highest maternal mortality rates, recommended that efforts to improve health facility readiness should be complemented with poverty reduction and educational interventions [52], which could be a lesson for further improving Nepal's situation.

The inventory/health worker interview questionnaire used in this analysis did not contain questions on the number of births and frequency of complications, which are important factors to consider in the assessment of the service readiness. Further, service availability and

readiness in facilities change periodically. Nevertheless, the current study provides a baseline status of BEmONC service availability and readiness in Nepal. In the future, this can be compared to monitor health facility readiness and trends similar to research conducted in rural Madagascar, in which the trends in maternal and child health care over different time periods were assessed [53]. This helps identify and timely correct the barriers of maternal, neonatal, and child health services coverage and quality. Thus, longitudinal studies are needed to identify the changes in availability and readiness to BEmONC over time. Further, NHFS 2015 did not collect data on health workers' performance by observation of obstetric service provision, which is an important aspect of quality of care.

## Conclusions

The readiness of BEmONC services was higher in public hospitals compared to private and peripheral public health facilities, indicating the need to improve the quality by increased service readiness in later facility types. Less than two-fifths of health facilities had the availability of the trained staff, and about one-fourth of health facilities have the availability of the guidelines. The overall readiness score of the health facility to provide BEmONC services was associated with service delivery staff, the service hours, and periodic review of maternal and newborn deaths. This study indicates the need to improve the quality of BEmONC services, by means of increasing the number of service delivery staff, service hours, and periodic review of maternal and newborn deaths in the pathway to meet the SDG targets.

## Supporting information

**S1 Table. Summary of tracer items of each domain and measurement procedure of BEmONC readiness scores.**
(DOCX)

## Acknowledgments

The authors would like to thank the United States Agency for International Development's Demographic Health Survey program for providing the datasets. We are also thankful to Elizabeth W. Perry of School of Public Health, Georgia State University for her feedback and editing in English language.

## Author Contributions

**Conceptualization:** Kiran Acharya, Raj Kumar Subedi.

**Formal analysis:** Kiran Acharya, Raj Kumar Subedi.

**Methodology:** Kiran Acharya, Raj Kumar Subedi, Sushma Dahal, Rajendra Karkee.

**Supervision:** Rajendra Karkee.

**Validation:** Kiran Acharya, Raj Kumar Subedi, Sushma Dahal, Rajendra Karkee.

**Writing – original draft:** Kiran Acharya.

**Writing – review & editing:** Sushma Dahal.

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
