## [Decision Letter · Decision Letter 0]

17 May 2021

PONE-D-21-14034

Basic emergency obstetric and newborn care service availability and readiness in Nepal: analysis of the 2015 Nepal Health Facility Survey

PLOS ONE

Dear Dr. Karkee,

Thank you for submitting your manuscript to PLOS ONE. After careful consideration, we feel that it has merit but does not fully meet PLOS ONE’s publication criteria as it currently stands. Therefore, we invite you to submit a revised version of the manuscript that addresses the points raised during the review process.

We look forward to receiving your revised manuscript.

Kind regards,

Calistus Wilunda, DrPH

Academic Editor

PLOS ONE

Journal Requirements:

Additional Editor Comments:

Line 210. “Willingness” may not be the right term in defining BEmONC services readiness.

In Table 5, indicate in the footnote what the coefficient means. You could as well type this directly in the column heading, instead of writing “coefficient”. Please report the point estimates with 95% CIs and not the SE, which is not easy to interpret intuitively.

Please correctly interpret the coefficients in the text. For example, these two statements are confusing: “peripheral facilities is 6.42% points lower than hospitals” and “Public facilities is 12.2 points better than private”. Remember linear regression the coefficient represents a change in the outcome on average for a unit change in the independent variable.

Please indicate which variables were included in multivariable analysis. What was the criteria for including the variables? Facility type determines “Number of delivery service staffs per facility” and “Number of delivery beds per facility. Thus, in looking at the effect of “Facility type” it is inappropriate to adjust for these two variables because they are mediators.

Did you look at factors such as staff training and competence in EmONC .

Some facilities, especially lower-level health facilities, may not have provided a signal function simply because there was no patient who needed the service. Was this and other reasons for not providing the service considered?

Please include a map showing the geographic distribution of the facilities by BEmONC status

One of the significant factors associated with service readiness is Health facility type. This has not been mentioned in the abstract.

Although the journal has no word limit, authors are encouraged to be concise. Please try to reduce the length of the manuscript to make it readable. The Introduction is unnecessarily lengthy.

Please revise the manuscript for English grammar.

Reviewers' comments:

Reviewer's Responses to Questions

**Comments to the Author**

1. Is the manuscript technically sound, and do the data support the conclusions?

Reviewer #1: Yes

Reviewer #2: Yes

2. Has the statistical analysis been performed appropriately and rigorously? 

Reviewer #1: Yes

Reviewer #2: Yes

3. Have the authors made all data underlying the findings in their manuscript fully available?

Reviewer #1: Yes

Reviewer #2: No

4. Is the manuscript presented in an intelligible fashion and written in standard English?

Reviewer #1: Yes

Reviewer #2: Yes

5. Review Comments to the Author

Reviewer #1: Pl explain full, dont use abbreviation when you write first time in the article, see abstract and revise.

Abstract: Correct spelling cross sectional in method section.

Overall need English grammatical correction

Results in abstract needs more clarity.

Pl support your findings with local literature and regional evidence like one study was conducted in Pakistan with similar kind of results, you can cite this article:

https://bmchealthservres.biomedcentral.com/articles/10.1186/s12913-019-4830-6

This survey was representative and how frequently it was conducted??

How about private sectors? These were included in the survey

These services are offered as per WHO recommendations?

Conclusion needs to be revised.

Reviewer #2: Thank you for inviting me to review this interesting orginal research based on the secondary data analysis of 2015 Nepal Health Facility Survey.

There are few comments and observation in the research done by Karkee and team.

1. Currently the maternity health services is severely disrupted by COVID-19 pandemic and all aspect of health service delivery is severely affected by the second wave in Nepal. Researchers needs to set a context that this study was done in prepandemic era and service readiness. The study is focusing on the service readiness and availability of BEOC. It would add value if the health workers performance is also added to it and provides the quality of care in BEOC, especially the signal functions.

2. There are some studies which has not be referenced well such as Thapa J et al, Equity and Coverage in the Continuum of Reproductive, Maternal, Newborn and Child Health Services in Nepal and Kruk et al. Mortality due to low-quality health systems in the universal health coverage era. Further how is this study so different from KC A et al Quality of Care for Maternal and Newborn Health in Health Facilities in Nepal.

Comments in the flow and content

Abstract- Introduction- Since, the paper is on readiness, so please revise the first two lines in the abstract intro. The result section doesnot provide findings, please provide numbers.

Background- The background is too long and doesnot flows well. Paragraphs 4-7, describes what progress has been made in maternal and newborn health, I think this is not the objective of the study. The researchers should focus on the definition of BEOC, when did this start in Nepal, what infra-structure is required.

Method- It doesnot follow the STROBE checklist. The method section is more in a report format, please provide the public engagement process in the development of research project.

Result section- There are too many tables, the most important table to describe is table 3-5. I suggest to remove table 1 (in method) and table 2.

Discussion- The first paragraph should be a summary of results, doesnot have so.

There needs to be a significant improvement, if academic editor deems it for further consideration.

6. PLOS authors have the option to publish the peer review history of their article (what does this mean?). If published, this will include your full peer review and any attached files.

Reviewer #1: **Yes: **Dr Ramesh Kumar, Professor Health Services Academy

Reviewer #2: No

---

## [Author Response · Author response to Decision Letter 0]

16 Jun 2021

Editorial Comments

R. We have revised the manuscript to confirm PLOS ONE’s style requirements.

Additional Editor Comments:

Line 210. “Willingness” may not be the right term in defining BEmONC services readiness.

R. Agreed and removed.

In Table 5, indicate in the footnote what the coefficient means. You could as well type this directly in the column heading, instead of writing “coefficient”. Please report the point estimates with 95% CIs and not the SE, which is not easy to interpret intuitively.Please correctly interpret the coefficients in the text. For example, these two statements are confusing: “peripheral facilities is 6.42% points lower than hospitals” and “Public facilities is 12.2 points better than private”. Remember linear regression the coefficient represents a change in the outcome on average for a unit change in the independent variable.

R. Thank you for the suggestion. We have added in the footnote what the coefficient means and have replaced SE with 95% CIs. We have also revised the text to make interpretation clear as follows:

Table 4 shows the results of the simple and multiple linear regression methods.Collectively, the variables included in the model explained 22.1% of the variation in BEmONC service readiness (R squared=0.221). The results of multiple linear regression analysis showed that the public facilities have higher readiness score (14.69 points, p<0.001) compared to the private facilities. Similarly, health facilities from Lumbini province had higher readiness scores (7.55 points, P<0.01) compared to those from Province 1. The readiness of the health facilities to provide BEmONC services was 4.89 points higher (P< 0.05) at facilities having 24 hour staffing than those that do not offer 24-hour services. The facilities that reported regular reviewing of maternal and newborn deaths had 4.88 points higher readiness (P<0.05) compared to facilities that did not report reviewing maternal and neonatal deaths. The service readiness in facilities increased 2.49 points (P<0.001) for each additional delivery staff person working at the facility.

Please indicate which variables were included in multivariable analysis. What was the criteria for including the variables? Facility type determines “Number of delivery service staffs per facility” and “Number of delivery beds per facility. Thus, in looking at the effect of “Facility type” it is inappropriate to adjust for these two variables because they are mediators.

R. We have indicated the independent variables that were included in multiple linear regression in Methods; Measurement of Variables:

“The independent variables included in the regression analysis were: facility type (hospitals or peripheral), ownership of facility (private or public), location of facility (rural or urban and by province), ecological region (Mountain, Hill and Terai), external supervision in the facility in last 4 months (occurred and not occurred), reviews of maternal or neonatal deaths (reviewed and not reviewed), system of determining and reviewing clients’ opinion (reviewed and not reviewed), quality assurance performed at least once a year (performed and not performed), duty schedule or call list for 24-hour staff assignment (Yes and No),the number of delivery service staff and the number of beds per facility. The peripheral health facilities include primary health care centres, health posts, and urban health centres.”

We selected these variables based on the available literature (26-29); and their importance in service readiness. We have checked the multicollineariety among these variables but did not find high correlation. Besides, number of delivery beds does not necessarily correlated with the number of staffs because many facilities often do not have adequate staff or staff absent.

Did you look at factors such as staff training and competence in EmONC. Some facilities, especially lower-level health facilities, may not have provided a signal function simply because there was no patient who needed the service. Was this and other reasons for not providing the service considered?

R. Thank you for the suggestion Staff training is one of the indicators in estimating the readiness score (see S1 Table). 

Yes, as you mentioned, some lower facilities might not have provided a signal function because there was no patient who needed the service. That is why, we included only those health facilities (n=457) that provide delivery and newborn care services. This has been mentioned in the sample and sampling procedure section

Please include a map showing the geographic distribution of the facilities by BEmONC status.

R. Included (Fig 1)

One of the significant factors associated with service readiness is Health facility type. This has not been mentioned in the abstract.

R We have revised the results in the abstract:

“The overall availability of the seven signal functions and BEmONC service readiness score was significantly higher in public hospitals compared with private hospitals and peripheral public health facilities. Significant factors associated with service readiness score were the ownership of facility (14.69 points higher in public hospitals, P<0.001), number of service delivery staff (2.49 points increase per each additional delivery staff, P<0.001), the service hours (4.89 points higher in facilities having 24 hours services, p< 0.05) and status of periodic review of maternal and neonatal deaths (4.88 points higher in facilities that conducted periodic review, p<0.05).”

Although the journal has no word limit, authors are encouraged to be concise. Please try to reduce the length of the manuscript to make it readable. The Introduction is unnecessarily lengthy.

Please revise the manuscript for English grammar.

R. We have got a native colleague edited this manuscript for grammar and have made it concise by shortening the introduction section.

Reviewers' comments:

Reviewer #1:

Pl explain full, dont use abbreviation when you write first time in the article, see abstract and revise.

R. Thank you for the suggestion, revised.

Abstract: Correct spelling cross sectional in method section.

Overall need English grammatical correction.

R. Done and edited for grammar by a native colleague.

Results in abstract needs more clarity.

R. Thank you. We have revised results section thoroughly. See the highlighted portion.

Pl support your findings with local literature and regional evidence like one study was conducted in Pakistan with similar kind of results, you can cite this article:https://bmchealthservres.biomedcentral.com/articles/10.1186/s12913-019-4830-6

R. Agreed and cited in discussion.

This survey was representative and how frequently it was conducted??

How about private sectors? These were included in the survey

R. We have mentioned in the study design section that this survey is the first comprehensive nationally representative cross-sectional health facility survey in Nepal. It was planned to conduct in every 5 years (however this will happen as per the need of Ministry of Health and Population). The second-round survey has been stopped currently in Nepal due to COVID-19 and our study can help to know the status before COVID-19.

Yes, the private hospitals were included in the survey; and we have indicated this in the sample and sampling procedure, Methods section. Indeed, one of the variable to distinguish the type of health facility is ‘ownership of facility’ in two category- public or private.

These services are offered as per WHO recommendations?

R. Yes

Conclusion needs to be revised.

R. Revised (see highlighted).

Reviewer #2:

Thank you for inviting me to review this interesting orginal research based on the secondary data analysis of 2015 Nepal Health Facility Survey.There are few comments and observation in the research done by Karkee and team.

1. Currently the maternity health services is severely disrupted by COVID-19 pandemic and all aspect of health service delivery is severely affected by the second wave in Nepal. Researchers needs to set a context that this study was done in prepandemic era and service readiness. 

R. Thank you for your suggestion. We hope that the title and objective will clarify the timeline of the assessment; and we have revised a sentence to indicate this (last paragraph, Introduction):

“Thus, we aim to assess the basic emergency obstetric and newborn care service availability and readiness among health facilities prior to the COVID-19 pandemic. in Nepal using nationally representative data from NHFS, 2015.”

The study is focusing on the service readiness and availability of BEOC. It would add value if the health workers performance is also added to it and provides the quality of care in BEOC, especially the signal functions.

R. Thank you for this comment and suggestion. As per our objectives, we focussed on the service readiness and availability; and service readiness was measured based on WHO SARA- manual, that do not include the work performance indicator. We have now indicated this in the limitation.

2. There are some studies which has not be referenced well such as Thapa J et al, Equity and Coverage in the Continuum of Reproductive, Maternal, Newborn and Child Health Services in Nepal and Kruk et al. Mortality due to low-quality health systems in the universal health coverage era. Further how is this study so different from KC A et al Quality of Care for Maternal and Newborn Health in Health Facilities in Nepal.

R. These references were cited in the introduction. Though the study from KC A et al derives data from same survey, the analysis approach and focus in our study differs; for example we have focussed to find out BEmONC service availability and readiness based on WHO SARA manual and have investigated factors affecting service readiness by multiple linear regression.

Comments in the flow and content

Abstract- Introduction- Since, the paper is on readiness, so please revise the first two lines in the abstract intro. The result section doesnot provide findings, please provide numbers.

R. We have revised the introduction and results section in the abstract as follows:

“Achieving maternal and newborn related Sustainable Development Goals targets is challenging for Nepal, mainly due to poor quality of maternity services.”

“The overall availability of the seven signal functions and BEmONC service readiness score was significantly higher in public hospitals compared with private hospitals and peripheral public health facilities. Significant factors associated with service readiness score were the ownership of facility (14.69 points higher in public hospitals, P<0.001), number of service delivery staff (2.49 points increase per each additional delivery staff, P<0.001), the service hours (4.89 points higher in facilities having 24 hours services, p< 0.05) and status of periodic review of maternal and neonatal deaths (4.88 points higher in facilities that conducted periodic review, p<0.05).”

Background- The background is too long and doesnot flows well. Paragraphs 4-7, describes what progress has been made in maternal and newborn health, I think this is not the objective of the study. The researchers should focus on the definition of BEOC, when did this start in Nepal, what infra-structure is required.

R. We have shortened the background section by revising, and deleting paragraphs 5 and 6 (see highlighted). We included definition of BEmONC in the ‘Introduction’ and infra-structure for it has been given in methods.

Method- It doesnot follow the STROBE checklist. The method section is more in a report format, please provide the public engagement process in the development of research project.

R. We have revised the methods section (See highlighted). There was not any specific public engagement process in the development of this survey research except involvement of multiple partners in the design and conduct of the survey, which has been mentioned in the methods, data sources.

Result section- There are too many tables, the most important table to describe is table 3-5. I suggest to remove table 1 (in method) and table 2.

R.We have moved Table 1 into supporting document but retained Table 2 as it provides important overview of included health facilities in this study.We have also refined other tables.

Discussion- The first paragraph should be a summary of results, doesnot have so.

R. We have revised the first paragraph of discussion as follows:

In this study, we assessed the availability and readiness of BEmONC services using data from NHFS, 2015. The overall availability of the seven signal functions and BEmONC service readiness score was significantly higher in public hospitals compared with private hospitals and peripheral public health facilities. Factors associated with service readiness were the ownership of facility, number of service delivery staff, the service hours and status of periodic review of maternal and neonatal deaths. 

There needs to be a significant improvement, if academic editor deems it for further consideration.

R. We have revised the manuscript thoroughly and have got a native colleague (Elizabeth W. Perry of School of Public Health, Georgia State University) edit English grammar.

---

## [Decision Letter · Decision Letter 1]

24 Jun 2021

PONE-D-21-14034R1

Basic emergency obstetric and newborn care service availability and readiness in Nepal: analysis of the 2015 Nepal Health Facility Survey

PLOS ONE

Dear Dr. Karkee,

Thank you for addressing the reviewers’ comments and for including a map of the health facilities. However, there are still minor errors to be addressed.

Line 26: edit this “We utilizedcross-sectionaldata...”

Line 282: Should be “the” instead of “thee”

Please use “Stata” instead of “STATA.”

Line 296: edit this “UnstandardizedCoefficientthat…”.

There are several places where you need to include a space between a full stop and the beginning of a new sentence (e.g. lines 77, 83, 93).

Please revise the legend of Figure 1 for English grammar.

Please indicate the exact p-value instead of p< 0.05. It is fine to write P<0.001.

This study did not assess the quality of care, thus, this statement “The qualities of emergency obstetric services need to be monitored in private hospitals” is not fully supported by the results. Moreover, it is not appropriate to say “qualities”. Please review this also in the conclusion of the main text. The basis for your conclusion on quality of care (which is a much broader concept) is not clear, although this may be a real problem.

You have acknowledged that you did not assess the quality of care (line 440), so addressing the additional comment raised by the second reviewer is at your discretion.  However, “… we did not analyze on process quality…” can be clarified. I guess you meant to say you did not assess the quality of care…

Therefore, we invite you to submit a revised version of the manuscript that addresses the above issues.

We look forward to receiving your revised manuscript.

Kind regards,

Calistus Wilunda, DrPH

Academic Editor

PLOS ONE

Journal Requirements:

Reviewers' comments:

Reviewer's Responses to Questions

**Comments to the Author**

1. If the authors have adequately addressed your comments raised in a previous round of review and you feel that this manuscript is now acceptable for publication, you may indicate that here to bypass the “Comments to the Author” section, enter your conflict of interest statement in the “Confidential to Editor” section, and submit your "Accept" recommendation.

Reviewer #1: All comments have been addressed

Reviewer #2: All comments have been addressed

2. Is the manuscript technically sound, and do the data support the conclusions?

Reviewer #1: Yes

Reviewer #2: Yes

3. Has the statistical analysis been performed appropriately and rigorously? 

Reviewer #1: Yes

Reviewer #2: Yes

4. Have the authors made all data underlying the findings in their manuscript fully available?

Reviewer #1: Yes

Reviewer #2: Yes

5. Is the manuscript presented in an intelligible fashion and written in standard English?

Reviewer #1: Yes

Reviewer #2: Yes

6. Review Comments to the Author

Reviewer #1: This is an important study in its filed. Author has addressed all the comments. This is an important study in its filed. Author has addressed all the comments.

Reviewer #2: Thank you for addressing the comments and observations made to your research work. The manuscript now is in an improved form. There is one last additional query that authors might want to address in the discussion section. since, there are studies been done from Nepal, especially by KC A et al (Perfect Storm, 2021) on quality of CEOC obstetric care varies with the number of childbirth in the health facilities. since, this study aim to assess the readiness of BEOC, can author put in the limitation that the quality of services to number of childbirth per health facility could not be assessed.

7. PLOS authors have the option to publish the peer review history of their article (what does this mean?). If published, this will include your full peer review and any attached files.

Reviewer #1: **Yes: **Ramesh Kumar

Reviewer #2: No

---

## [Author Response · Author response to Decision Letter 1]

28 Jun 2021

Line 26: edit this “We utilizedcross-sectionaldata...”

Line 282: Should be “the” instead of “thee”

Please use “Stata” instead of “STATA.”

Line 296: edit this “UnstandardizedCoefficientthat…”.

There are several places where you need to include a space between a full stop and the beginning of a new sentence (e.g. lines 77, 83, 93).

R. We edited these typing mistakes. Thank you for highlighting these.

Please revise the legend of Figure 1 for English grammar.

R. Revised: BEmONC service readiness index in health facilities by province-wise

Please indicate the exact p-value instead of p< 0.05. It is fine to write P<0.001.

R. Indicated. We also added a column of P value in Table 4.

This study did not assess the quality of care, thus, this statement “The qualities of emergency obstetric services need to be monitored in private hospitals” is not fully supported by the results. Moreover, it is not appropriate to say “qualities”. Please review this also in the conclusion of the main text. The basis for your conclusion on quality of care (which is a much broader concept) is not clear, although this may be a real problem.

R. The statement was revised: 

“The private hospitals need to be encouraged for BEmONC service readiness.” in conclusion of abstract.

“The availability of the seven signal functions and readiness of BEmONC services was higher in public hospitals compared to private and peripheral public health facilities, indicating the need to improve the quality by increased service readiness in later facility types.” In conclusion of main text.

You have acknowledged that you did not assess the quality of care (line 440), so addressing the additional comment raised by the second reviewer is at your discretion. However, “… we did not analyze on process quality…” can be clarified. I guess you meant to say you did not assess the quality of care…

R. We revised the sentence to make it more clear: “Further, NHFS 2015 did not collect data on health workers’ performance by observation of obstetric service provision, which is an important aspect of quality of care.”

---

## [Editor Report · Decision Letter 2]

30 Jun 2021

Basic emergency obstetric and newborn care service availability and readiness in Nepal: analysis of the 2015 Nepal Health Facility Survey

PONE-D-21-14034R2

Dear Dr. Karkee,

We’re pleased to inform you that your manuscript has been judged scientifically suitable for publication and will be formally accepted for publication once it meets all outstanding technical requirements.

Kind regards,

Calistus Wilunda, DrPH

Academic Editor

PLOS ONE

Additional Editor Comments (optional):

Stata is not an abbreviation. Please change "STATA" to “Stata” when you receive an email to make amendments.
---

## [Editor Report · Acceptance letter]

7 Jul 2021

PONE-D-21-14034R2 

Basic emergency obstetric and newborn care service availability and readiness in Nepal: analysis of the 2015 Nepal Health Facility Survey 

Dear Dr. Karkee:

I'm pleased to inform you that your manuscript has been deemed suitable for publication in PLOS ONE. Congratulations! Your manuscript is now with our production department. 

Kind regards, 

on behalf of

Dr. Calistus Wilunda 

Academic Editor

PLOS ONE